# Electrical Re-Writable Non-Volatile Memory Device Based on PEDOT:PSS Thin Film

**DOI:** 10.3390/mi11020182

**Published:** 2020-02-10

**Authors:** Iulia Salaoru, Christos Christodoulos Pantelidis

**Affiliations:** Emerging Technologies Research Centre, De Montfort University, The Gateway, Leicester LE1 9BH, UK

**Keywords:** PEDOT:PSS, memory cells, bistability, retention time, switching

## Abstract

In this research, we investigate the memory behavior of poly(3,4 ethylenedioxythiophene) polystyrene sulfonate (PEDOT:PSS) cross bar structure memory cells. We demonstrate that Al/PEDOT:PSS/Al cells fabricated elements exhibit a bipolar switching and reproducible behavior via current–voltage, endurance, and retention time tests. We ascribe the physical origin of the bipolar switching to the change of the electrical conductivity of PEDOT:PSS due to electrical field induced dipolar reorientation.

## 1. Introduction

Today, electronic memory elements are an essential component of all electronic devices, from computers to toys and from health monitors to space technology. This is why these devices attract a huge interest from both researchers and the industry community. Additionally, the current trend in electronics is shifting from rigid to flexible, bendable electronic components and devices. In this context of re-defining the memory technology, organic materials are the best candidates to fulfill the current pathway in memory technologies and applications. It should be highlighted that organic materials offer a large number of advantages, for example, they can be processed easily at low temperatures (room temperature) over a large area on flexible substrates using wet-processing techniques, have a low weight, great mechanical flexibility, and a very important aspect is that the electrical and optical properties can be easily tailored. Due to all above mentioned advantages, the organic materials are being explored for a wide variety of electronic devices from light emitted diode to photovoltaics, batteries, transistors, and memories. The organic memory devices can be fabricated either by depositing a polymer and/or a blend of polymer and small organic molecules or nanoparticles between two metal electrodes. The role of the organic materials in the field of memories have been highlighted in a handful of comprehensive review papers [1,2,3,4]. 

In this research, we will focus on memory devices based on polyelectrolyte complexes, i.e., poly(3,4-ethylenedioxythiophene) polystyrene sulfonate (PEDOT:PSS). Indeed, Nguyen and Lee [5] investigated the memory behaviour of PEDOT:PSS planar configuration with a gold (Au) bottom electrode and used gold probe as top contact. The authors reported that two memory effects coexisted in those cells, firstly a WORM (write once read many times) and secondly a switchable diode effect in via charge trapping and cation movement [5]. On the other side, Liu et al. [6] fabricated a fully Au/PEDOT:PSS/Au structure where a nonpolar behavior has been observed. The authors proposed a possible mechanism for the observed nonpolar behavior, filamentary formation by oxidation, and reduction of the PEDOT:PSS. Moreover, further suggestions, i.e., diffusion of Au atoms into an active core during the fabrication procedure have been proposed. On another side, Jeong et al. [7] presented a bipolar switching operation of devices based on PEDOT:PSS (70 nm thickness) with Al as bottom and top electrodes. In this case, the authors suggested that both interfaces play a key role in the switching behavior either through redox reaction between the top Al electrode and the active core and the Al native oxide formed at the bottom side. Wang et al. [8] fabricated and tested AI/PEDOT:PSS/Al as well and they proposed the formation and rupture of conductive filaments being responsible for the bistability behavior; a total different mechanism as proposed in [7] with even the same metal (Al) used as top and bottom electrode and the same thickness of the active core (70 nm). Interestingly, a unipolar switching in Al/PEDOT:PSS (70 nm)/Al has been presented by Kim and Kim [9]. The authors proposed thermally (Joule effect) driven mechanisms based on formation and rupture of the conductive filament across the cell. Furthermore, they validated that there is not any influence/formation of the oxide on the top of the bottom electrode by replacing Al with a noble Au metal. Indium tin oxide (ITO)/PEDOT:PSS/Al structure was fabricated and investigated by Ha and Kim [10] and these devices exhibit bipolar switching characteristics. The authors suggested that the mechanism responsible for the observed behavior is based on formation and destruction of the current paths through reduction and oxidation reactions of the PEDOT chain in PEDOT:PSS thin films. Furthermore, the effect of various metals as both bottom and top electrodes on the switching behavior have been studied by Ha and Kim [11]. The transparent conducting oxide, indium tin oxide (ITO), and Al were used as bottom electrode, and Al, Ti, Cr, Pt, Ni, ITO, Pd, and Au were used for the top electrode (TE). The authors reported that the only devices that exhibited bipolar switching behavior were based on ITO/PEDOT:PSS/TE, whilst the Al/PEDOT:PSS/TE devices showed unipolar switching behavior but only under the control of compliance current. As it can be seen from Figure 1, there is an inconsistency between reports (where some groups observed bipolar switching and others unipolar switching even using the same structure and materials on the same systems), confusing the academic society. Indeed, some possible mechanisms for the switching behaviour in PEDOT:PSS memory devices have been reported. However, these proposed mechanisms are mainly focused on the interface interaction and effect of the metal electrodes than the intrinsic properties of an active core polymer mixture. 

In this paper, we report a bipolar switching and reproducible behaviour in PEDOT:PSS-based cross bar structure memory cells. The memory capacity of the fabricated cells was elucidated through comprehensive electrical tests, i.e., current–voltage characteristics, endurance, and retention time. To date, the majority of the mechanisms presented in literature are neglecting the changes in the electrical conductivity of PEDOT:PSS due to the conformational changes under an electrical field. Here, we consider these changes and we ascribe the physical origin of the bipolar switching to the change of the electrical conductivity of PEDOT:PSS due to electrical field induced dipolar reorientation.

## 2. Experimental

In this study, we used PEDOT:PSS as active core of the memory cells. The PEDOT:PSS was purchased from Sigma Aldrich with a concentration of 2% PEDOT:PSS in ethylene glycol monobutyl ether/water (3:2). For electrical measurements (current–voltage (I–V) characteristics, endurance test, and retention time), metal/polymer/metal (MPM) structures were fabricated. In order to fabricate MPM structures, firstly the conductive tracks of aluminium (Al) were deposited by thermal evaporation (deposition rate was about 1 nm/s) on a glass substrate to define the bottom electrode (BE) of the final devices. Then, the PEDOT:PSS active layer was deposited by spin coating, with a spin coating speed of 3000 rpm, onto a glass substrate marked with Al tracks. The film thickness was 57 nm and was measured using a Rudolf AutoEL III ellipsometer. The PEDOT:PSS film was dried in air without any post-deposition heat treatment. The electrical resisitivity of the PEDOT:PSS deposited layer was around 850 Ω·cm and was tested usind 4 point probe measurements. Finally, in order to achieve a cross bar architecture, the top electrode (TE), Al, was deposited by thermal evaporation. The electrical measurements were performed using a two-probe system, the probes used Karl Suss PH100 probe heads connected to a PC driven picoammeter (HP4140B). The current–voltage (I–V), endurance and retention time characteristics of the memory device structures were measured in a Faraday cage in the dark at room temperature. 

## 3. Results and Discussion 

The memory cells investigated here are based on a Al (100 nm) top electrode (TE)/PEDOT:PSS active layer (57 nm)/Al (70 nm) bottom electrode stack fabricated on a glass substrate as schematically represented in Figure 2a. Figure 2b shows the optical image of the single device viewed from the TE with its active junction of 1 mm × 1 mm. 

The devices were first electrically characterized via voltage sweeping with the results presented in the Figure 3. Electrical biasing of the device was carried out both for assessing and modifying its resistive states. In order to identify the appropriate voltage to switch the devices as well as to ensure that the devices do not exhibit an electrical breakdown, the device was biased via three consecutive switching cycles, in steps of 100 mV, commencing from 0 V that follow the sequence positive–negative in the applied potential. In Figure 3a, a first cycle, represented by the black color curve, where the voltage follows the sequence 0 V/+2 V back to 0V, and then −2 V, and finally back to 0 V, a second cycle, represented by the blue color curve, where the voltage was changed from 4 V to −4 V, and the last cycle represented by the red color curve where a maximum of +6/−6 V is applied. In the third cycle, the device exhibited bistable I–V behavior (Figure 3b) with the device switching from a high conductivity state (HCS) to a low conductivity state (LCS) at 6 V. Resetting the device to HCS was achieved by a negative potential of −6 V. Additionally, the Vset, Vreset, and Vread were identified from the third cycle and those values were further used when memory retention time and endurance tests were performed. Furthermore, we tested all 16 devices from the same sample with 13 of them having a reliable behavior as can be seen in Figure 3c.

We further proceed our study by investigating the endurance (write/read/erase/read) capacity of the PEDOT:PSS-based memory cells over 1000 pulses. The device’s endurance behavior was tested by applying a write voltage of 6 V that is setting the device in LCS, then the read voltage of 4 V was applied in order to read this state. Switching to HCS was performed by applying a voltage pulse of −6 V and reading took place at 4 V. The endurance results and the employed voltage scheme are presented in Figure 4. 

Memory retention time was also investigated for the Al/PEDOT:PSS/Al memory elements. The low conductivity state was programmed by applying a pulse of an amplitude of +6 V and 1 s width, and then the state was read and the current monitored by bringing the voltage down to 4 V. Then, by applying a pulse of −6 V amplitude/1 s width, the device was switched to a high conductivity state and the state was read at 4 V. The high and low conductivity states remain distinguishable as can be seen from Figure 5. 

We suggest that the switching mechanism, in PEDOT: PSS based memory elements, is due to the electrical field induced dipole formation/reorientation (conformational change) that creates two distinctive conductive states. We previously demonstrated that formation/orientation of the dipole structure within the active core, under the influence of external electrical field, led to change in the conductivity, hence creating two distinctive electrical states that actually define a memory behavior [12,13,14,15,16]. Systems as admixture of polystyrene (PS), fullerene (C60) and 8-Hydroxyquinoline (8HQ) [12], polyvinyl acetate (PVAc) and sea salt (NaCl) [13], blend of polyvinyl acetate (PVAc) and barium titanate (BaTiO_3_) nanoparticles [14], polyvinyl acetate (PVAc) and blend of two small organic molecules tetracyanoethylene (TCNE) and tetrathiafulvalene (TTF) [16] have been investigated and the results fully supported the proposed model, i.e., formation/orientation of the dipoles under the action of the external electric field. As PEDOT:PSS is a polymer mixture of PSS (sulfonated polystyrene) that is negatively charged and PEDOT (poly(3,4-ethylenedioxythiophene) that is positively charged, a dipole structure is expected to be present in this polymer blend. In the absence of the external electric field, these electrical dipoles are randomly orientated. 

When the external electrical field (SET) was applied, the dipoles aligned along the electrical field, which determined the rise of the internal electric field. Next, the applied pulse was read pulse. As can be seen from Figure 6b, the internal electric field and read potential are orientated oppositely. In this situation, the effective electric field across the device was reduced and thus the device exhibited a low conductivity state (LCS). On the other hand, the high conductivity state was achieved when the internal electric field had the same direction with the reading one and this occurred when the RESET pulse was applied to the cell (Figure 6c). In summary, the internal electric field can either enhance or diminishe the external electric field applied to the device, and hence generate two conductivity states, i.e., low and high states. 

The approach presented in this paper, based on electrical field dipolar reorientation, is supported by a study published by Mahajan et al. [17] where it has been demonstrated that changes in the electrical conductivity of PEDOT:PSS thin film due to the electrical field induced dipolar reorientation. 

## 4. Conclusions

In summary, we have studied the memory capabilities of Al/PEDOT:PSS/Al crossbar-type array cells. All electrical tests performed (I–V characteristics, endurance, and retention time) clearly demonstrated that PEDOT:PSS-based solid state devices display memory behavior. We suggest that the dipole reorientation induced by an external electric field is responsible for the creation of two distinctive electrical states. This further strengthens our previously proposed working mechanism, i.e., the switching between a high- and a low-conductivity states is determined entirely by the dipole formation and their rotation with applied electric field. In a nutshell, these results corroborate with both, our previous studies [13,14,15,16] and the model proposed by Paul [12] on the “Creation of electric dipoles or internal electric”, and fully support the conclusion that the aforementioned model is applicable to all systems that can develop an internal electric dipole/field. 

## Figures and Tables

**Figure 1 micromachines-11-00182-f001:**
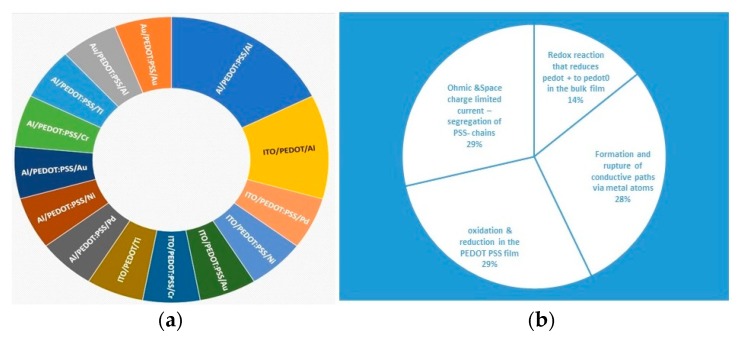
Poly(3,4 ethylenedioxythiophene) polystyrene sulfonate (PEDOT:PSS) based memory elements (**a**) metal/PEDOT:PSS/metal structures and (**b**) switching mechanisms reported in the literature [5,6,7,8,9,10,11].

**Figure 2 micromachines-11-00182-f002:**
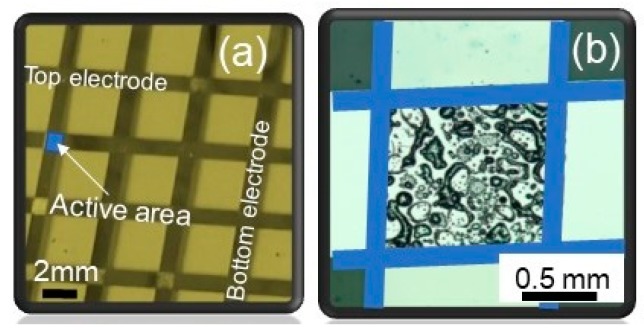
Optical microscope image of (**a**) crossbar type array consisting of devices with active area of 1 mm × 1 mm, (**b**) optical microscope image of a single 1 mm × 1 mm device.

**Figure 3 micromachines-11-00182-f003:**
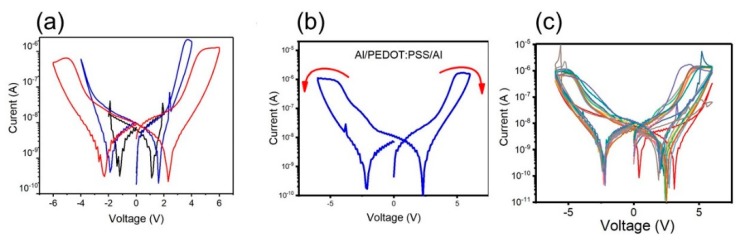
(**a**) Current–voltage (I–V) characteristic of three consecutive cycles of Al/PEDOT:PSS/Al memory elements; (**b**) I–V characteristics of one of the switched devices; (**c**) I–V characteristics of the 13 tested devices.

**Figure 4 micromachines-11-00182-f004:**
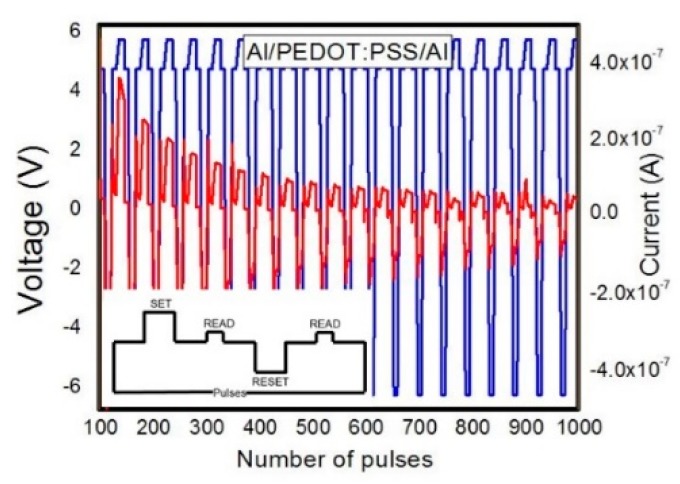
Endurance characteristic of Al/PEDOT:PSS/Al; the inserted picture depicts the pulsing sequences utilized for write/read/erase/read test.

**Figure 5 micromachines-11-00182-f005:**
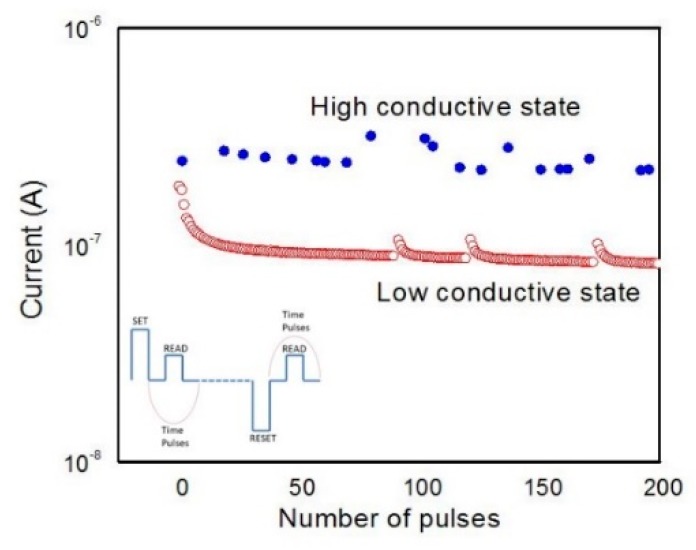
Memory retention time of PEDOT:PSS based memory cells (inserted picture depicts the pulsing sequences utilized for retention time test).

**Figure 6 micromachines-11-00182-f006:**
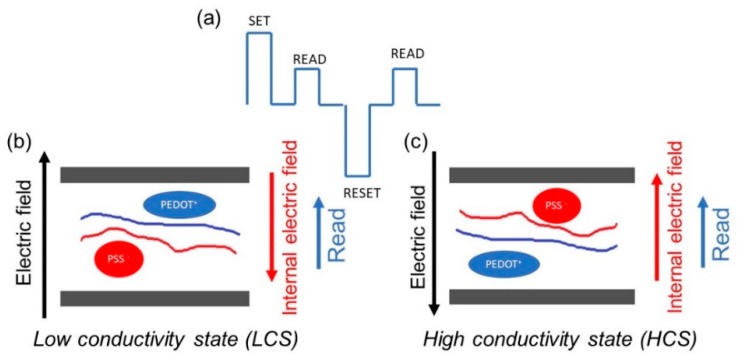
(**a**) Schematic diagrams explaining the memory mechanism in PEDOT:PSS-based memory cells; The device exhibited a (**b**) low conductivity state (LCS) and (**c**) high conductivity state (LCS).

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
