# Peer review of "Electrical Re-Writable Non-Volatile Memory Device Based on PEDOT:PSS Thin Film"

_micromachines, 2020, doi:10.3390/mi11020182_

Round 1

Reviewer 1 Report

In this manuscript the authors report on a two-terminal memory in the form of Al/PEDOT:PSS/Al. At the introduction, authors present the the wide range of interpretation of the memory phenomena that exist in the literature, despite the simplicity of the system. The results are relatively clear and the measurements are convincing. I therefore believe that the manuscript can be published in this journal. 

However, I believe that the interpretation of the results can be significantly improved and I suggest that the authors should address some of those aspects in a future work. For example:

It is known that there are several versions of PEDOT:PSS with different electronic properties and conductivity. Also post-deposition treatment of the polymer is crucial.  It is known that even from synthesis, these polymers have residual ionic species which may contribute or even dominate the memory phenomena. In order to be sure that the main mechanism is dipolar orientation, the authors should conduct additional experiments: a) temperature dependence IV scans. b) Study the glass transition temperature of the polymer to evaluate its molecular mobility at room temperature. 

Many inconsistencies in the literature stem from the fact that some of the above mentioned issues are not systematically studied or excluded.  

Author Response

We wish to thank this referee for the provided comments and suggestions and definitely we will perform the recommended experiments and we will publish the results and findings in the further publication(s).  

Reviewer 2 Report

Really nice work, congratulations ! 

But I have a few comments on this work; 

I have highlighted a few english editing issues that should be re-examined in ' '  : 

line 19 : 'health' monitors ? heat monitors ? please examine

line 26 : flexibility and 'a' very

line 39-40 : Liu 'et al' 

line 43 : procedure 'have' been proposed

line 59 : thin 'films' ?

line 69 : are mainly 'focused'

line 105 : that the devices 'don't exhibit a physical breakdown' ? please examine

      2. Some minor scientific comments 

line 34 : please replace ' polymer mixture of two ionomers' with : polyelectrolyte complexes 

line 65 and Figure 1 : 'conflicting findings' is a rather aggressive term - please redefine. A suggestion would be : 'there is an inconsistency between reports on the same systems, leading in  confusion for the academic society' Also, please re-make figure 1 so that the words on the right part don't fall into the lines. 

lines 82 - 89 : Please provide information on the metal deposition rates , on PEDOT:PSS spin coating speed and annealing temperature. 
Also, please report the electrical conductivity of the PEDOT:PSS systems that you used through a 4point probe measurement. 
This is an important detail on the reproducibility aspect of the PEDOT systems. 
How did you measure the thicknesses of your layers ?

Figure 2: Provide magnification details of optical microscope image on the image (i.e. as you have the pixel, have a double arrow to showcase the 1mm x 1mm) . If you could also provide a more homogenous image for Figure 2b, it would be great. 

lines 143 -147 : How does this current study compare with the previous ones that you mention ? 

Good luck !

Author Response

We wish to thank this referee for the provided comments and for appreciation of our work.

line 19 : 'health' monitors ? heat monitors ? please examine

Nowadays, most of the heath monitors such as blood glucose meter, blood pressure monitor have incorporated memory in order to store the readings.

line 26 : flexibility and 'a' very

Amended accordingly

line 39-40 : Liu 'et al' 

Amended accordingly

line 43 : procedure 'have' been proposed

Amended accordingly

line 59 : thin 'films' ?

Amended accordingly

line 69 : are mainly 'focused'

Amended accordingly

line 105 : that the devices 'don't exhibit a physical breakdown' ? please examine

replace with “don’t exhibit an electrical breakdown”

Some minor scientific comments 

line 34 : please replace ' polymer mixture of two ionomers' with : polyelectrolyte complexes 

Amended as suggested

line 65 and Figure 1 : 'conflicting findings' is a rather aggressive term - please redefine. A suggestion would be : 'there is an inconsistency between reports on the same systems, leading in  confusion for the academic society' Also, please re-make figure 1 so that the words on the right part don't fall into the lines. 

Thank you very much for this suggestion. Below is the amended text:

“there is an inconsistency between reports (where some groups observed bipolar switching and others unipolar switching even using the same structure and materials on the same systems), leading in confusion for the academic society”

lines 82 - 89 : Please provide information on the metal deposition rates , on PEDOT:PSS spin coating speed and annealing temperature. 
Also, please report the electrical conductivity of the PEDOT:PSS systems that you used through a 4point probe measurement. 
This is an important detail on the reproducibility aspect of the PEDOT systems. 
How did you measure the thicknesses of your layers?

The requested information (metal deposition rates, PEDOT:PSS spin coating speed and annealing temperature, method used to measure the thickness, electrical resisitivity) were added to the revised manuscript

Figure 2: Provide magnification details of optical microscope image on the image (i.e. as you have the pixel, have a double arrow to showcase the 1mm x 1mm). If you could also provide a more homogenous image for Figure 2b, it would be great. 

Amended accordingly

lines 143 -147 : How does this current study compare with the previous ones that you mention ? 

In a nutshell, these results corroborate with both, our previously studies [13-16] and the model proposed by Paul, S [12] on the “Creation of electric dipoles or internal electric” fully support the conclusion that the aforementioned model is applicable to all systems that can develop an internal electric dipole/field.
